# Diversity and Distribution of *β*-Lactamase Genes Circulating in Indian Isolates of Multidrug-Resistant *Klebsiella pneumoniae*

**DOI:** 10.3390/antibiotics12030449

**Published:** 2023-02-23

**Authors:** Suraj Shukla, Siddhi Desai, Ashutosh Bagchi, Pushpendra Singh, Madhvi Joshi, Chaitanya Joshi, Jyoti Patankar, Geeti Maheshwari, Ekadashi Rajni, Manali Shah, Devarshi Gajjar

**Affiliations:** 1Department of Microbiology and Biotechnology Centre, Faculty of Science, The Maharaja Sayajirao University of Baroda, Vadodara 390002, Gujarat, India; 2Amity Institute of Biotechnology, Amity University of Noida, Noida 201313, Uttar Pradesh, India; 3ICMR-National Institute of Research in Tribal Health, Jabalpur 482003, Madhya Pradesh, India; 4Gujarat Biotechnology Research Centre, Department of Science and Technology, Government of Gujarat, Gandhinagar 382011, Gujarat, India; 5Sterling Hospital, Vadodara 390007, Gujarat, India; 6Toprani Advanced Lab Systems, Vadodara 390020, Gujarat, India; 7Department of Microbiology, Mahatma Gandhi University of Medical Sciences & Technology, Jaipur 302015, Rajasthan, India; 8Desai Metropolis Health Service Pvt. Ltd., Surat 395001, Gujarat, India

**Keywords:** whole genome sequencing, *β*-lactamases, MLST, plasmid replicons, *Klebsiella pneumoniae*

## Abstract

*Klebsiella pneumoniae* (Kp) has gained prominence in the last two decades due to its global spread as a multidrug-resistant (MDR) pathogen. Further, carbapenem-resistant Kp are emerging at an alarming rate. The objective of this study was (1) to evaluate the prevalence of *β*-lactamases, especially carbapenemases, in Kp isolates from India, and (2) determine the most prevalent sequence type (ST) and plasmids, and their association with *β*-lactamases. Clinical samples of *K. pneumoniae* (*n* = 65) were collected from various pathology labs, and drug susceptibility and minimum inhibitory concentrations (MIC) were detected. Whole genome sequencing (WGS) was performed for *n* = 22 resistant isolates, including multidrug-resistant (MDR) (*n* = 4), extensively drug-resistant (XDR) (*n* = 15), and pandrug-resistant (PDR) (*n* = 3) categories, and genomic analysis was performed using various bioinformatics tools. Additional Indian MDRKp genomes (*n* = 187) were retrieved using the Pathosystems Resource Integration Center (PATRIC) database. Detection of *β*-lactamase genes, location (on chromosome or plasmid), plasmid replicons, and ST of genomes was carried out using CARD, mlplasmids, PlasmidFinder, and PubMLST, respectively. All data were analyzed and summarized using the iTOL tool. ST231 was highest, followed by ST147, ST2096, and ST14, among Indian isolates. *bla*_ampH_ was detected as the most prevalent gene, followed by *bla*_CTX-M-15_ and *bla*_TEM-1_. Among carbapenemase genes, *bla*_OXA-232_ was prevalent and associated with ST231, ST2096, and ST14, which was followed by *bla*_NDM-5_, which was observed to be prevalent in ST147, ST395, and ST437. ST231 genomes were most commonly found to carry Col440I and ColKP3 plasmids. ST16 carried mainly ColKP3, and Col(BS512) was abundantly present in ST147 genomes. One Kp isolate with a novel MLST profile was identified, which carried *bla*_CTX-M-15_*, bla*_OXA-1_, and *bla*_TEM-1_. ST16 and ST14 are mostly dual-producers of carbapenem and ESBL genes and could be emerging high-risk clones in India.

## 1. Introduction

*Klebsiella pneumoniae* (Kp), a member of the *Enterobacteriaceae* family, is one of the commensal organisms in the gastrointestinal tract of healthy humans and animals [1]. Since the last two decades, Kp has gained importance because of its worldwide spread as a multidrug-resistant (MDR) pathogen. Further, Kp poses a great concern since the acquisition of plasmids and transposons carrying antibiotic resistance genes are not only restricted to horizontal transfer to other Kp strains but also other enteric bacteria [1]. The broad host range plasmids (IncX3, IncA/C, IncN, and IncL) acquired by Kp have made it the harbinger of resistance determinants among other enterobacteria [2]. Consequently, Kp was declared as a “Priority 1: CRITICAL” pathogen by the World Health Organization (WHO) in 2017 [3], while it was listed in the Indian priority pathogen list (IPPL) in 2021 [4], due to the increasing antibiotic resistance (ABR), including last-resort antibiotics such as carbapenems, colistin, and tigecycline. Further, carbapenem-resistant *Enterobacteriaceae* (CRE) are emerging at an alarming rate; hence, surveillance studies of MDRKp have become highly important.

The epidemiological features of carbapenem-resistant Kp have been previously reviewed [5]. The global presence of carbapenem resistance in Kp is mainly due to the presence of isolates containing class A-type *β*-lactamase (*bla*_KPC_), class B-type metallo-*β*-lactamase (*bla*_NDM_), and class D-type oxacillinase (*bla*_OXA-48_), with geographical variation. For example, Greece, Taiwan, Columbia, the USA, Canada, and China have many more strains that produce *bla*_KPC_ and *bla*_NDM_ than strains that produce *bla*_OXA-48_, which are far less common in those nations. While in the Arabian Peninsula and India, *bla*_OXA-48_ and *bla*_NDM_ producers are common while *bla*_KPC_ producers are rare.

The resistance rate against carbapenems and extended-spectrum *β*-lactamases (ESBLs) has been worryingly increasing in India in the past few years. A report from India has manifested an increase in carbapenem resistance rates from 9% in 2008 to 44% in 2010 [6]. Another report from a tertiary hospital in India reported 24.6% resistance to ESBLs in 2007 [7]. In 2017, carbapenem resistance was reported as high as 44% in India [8]. In 2018, a significant rise in the resistance level of ESBLs (45.1–93.1%) was reported in India, and the highest resistance of 84.9% was reported against cephalosporins [9]. A report from North India suggested a 29.4% resistance rate against carbapenems [10]. Recently, the overall prevalence of multidrug resistance (MDR: isolate that has developed resistance to at least one antimicrobial agent from three or more antimicrobial categories [11]) among Indian Kp isolates was reported at 58.0%. Further, it was also reported that tigecycline and colistin were the most effective drugs so far [12]. However, extensively drug-resistant (XDR: resistant to at least one agent in all antimicrobial groups except two or fewer, i.e., bacterial isolates remain susceptible to only one or two categories [11]) isolates with co-resistance against carbapenem and colistin [8,13] and pandrug-resistant (PDR: resistant to any antimicrobial agent [11]) isolates have also recently been reported in India [14].

Genome analysis of multidrug-resistant organisms provides us with important information about the phylogroup and the number of genes and plasmids in MDR bacteria at the same time. The multi-locus sequence typing (MLST) aids in identifying region-specific sequence types (STs) and their association with AMR genes. In recent times, extensive sequencing of pure genomes of Kp has led to huge momentum in providing a consolidated snapshot of resistance. Here, we report a comparison of all whole-genome sequences of Kp genomes from India. We leveraged a collection of 209 genomes including genomes of our 22 isolates available in the PATRIC database (https://www.patricbrc.org, accessed on 25 September 2021) and reported the most prevalent MLST, plasmid replicons, and AMR genes and their location in genomes for Indian isolates.

## 2. Results

We collected Kp isolates (*n* = 65) from Gujarat, of which *n* = 22 isolates were found to be resistant to multiple antimicrobial agents of various classes, and these were used for WGS analysis. Among the total 22 isolates, *n* = 3, *n* = 15, and *n* = 4 isolates were identified as PDR, XDR, and MDR, respectively (Table 1). PDR strains (*SBS12*, *DGL5*, *DGL8*) were resistant to all antimicrobial agents in all classes. All PDR isolates were resistant to colistin and tigecycline, with MIC values of ≥4 µg/mL and ≥2 µg/mL, respectively. Majority of the XDR isolates (*DGL6*, *DGL7*, *DGL9*, *DGL10*, *DGL11*, *DGGL12*, *DGL13*, *DGL14*, *DGL15*, *DGL16*, *DGL17*) were resistant to all antibiotics, including tigecycline, with a MIC of ≥2 µg/mL, and only remained susceptible against colistin. Two XDR isolates (*SBS3*, *SBS5*) were resistant to all antimicrobial agents except for colistin and tigecycline (Appendix A).

### 2.1. Distribution of Sequence Types (STs)

We investigated the genomic diversity of *Klebsiella pneumoniae* genomes circulating in the Indian subcontinent using genomes (*n* = 209) reported from 2012 to early 2021 (Figure 1). The analysis revealed that 50 different sequence types were circulating in India, of which ST231 (33.49%, *n* = 70) is the most prevalent, followed by ST147 (10.04%, *n* = 21), ST2096 (7.17%, *n* = 15), and ST14 (5.74%, *n* = 12). Other STs, such as ST43, ST395, ST16, ST11, ST15, ST23, ST35, ST48, ST101, ST307, ST437, ST515, ST420, and ST42, were found in at least 2 genomes, and 32 different STs were found individually in single genomes. SBS4 had unique combinations of 7 housekeeping genes (*gapA:198*, *infB:1*, *mdh:2*, *pgi:1*, *phoE:10*, *rpoB:4*, *tonB:764*), which was submitted to the Institut Pasteur MLST database and identified as novel ST5438. Regarding the origin of the genomes, majority of the genomes have human origin (92.82%, *n* = 194), of which 49.28% (*n* = 103) of isolates belonged to blood stream infections (BSIs), followed by respiratory tract infections (RTIs) (27.75%, *n* = 58), urinary tract infections (UTIs) (8.61%, *n* = 18), and the central nervous system (CNS) (0.95%, *n* = 2), while a single isolate was found for each infection, such as nosocomial, wound, sepsis, and implant infection. Here, 6.69% were of environmental origin (*n* = 14). The origins for 9 genomes were not found and these are mentioned in Figure 1 as not available.

Regarding the state-wise distribution of *Klebsiella* genomes, the highest number of genomes (63.15%, *n* = 132) was reported from the southern states of India (Tamil Nadu *n* = 128 and Kerala *n* = 4), followed by 15.13% from Eastern states (*n* = 32) (West Bengal *n* = 21, Assam *n* = 10, and Odisha *n* = 1), 12.44% from Western states (*n* = 26) (Gujarat *n* = 22 and Maharashtra *n* = 4), 7.17% from Northern states (*n* = 15) (New Delhi *n* = 11, Uttar Pradesh *n* = 2, and Punjab *n* = 2), and 1.43% from the central part of India (*n* = 3) (Madhya Pradesh) (Figure 2). The geographic location for one isolate (Strain: ME30) was not available. ST231 was observed to be the most prevalent ST among all regions of India. ST147 was detected in Tamil Nadu, Gujarat, West Bengal, and Uttar Pradesh, while ST2096, ST16, and ST42 were detected in Tamil Nadu and Gujarat. Presence of ST14 was detected in Tamil Nadu, West Bengal, Gujarat, and New Delhi, whereas ST307 was only found in Assam. State-wise distribution for STs is shown in Figure 2.

### 2.2. Distribution of Carbapenemases

The percentage of strains harboring the *β*-lactamase genes and categorization of *β*-lactamase genes (carbapenemases, ESBLs, BSBLs, and other *β*-lactamases) was performed based on their functional properties available in the Beta-Lactamase Database (http://bldb.eu/, accessed on 15 May 2022) (Figure 3). Distribution of carbapenemase genes was observed to vary with different STs (Figure 1 and Figure 4). The carbapenemase genes, *bla*_OXA-48-like_ (*bla*_OXA-48_, *bla*_OXA-232_, and *bla*_OXA-181_), were found to be the most circulating among Indian isolates, where *bla*_OXA-232_ was the most prevalent carbapenemase, found in 46.41% (*n* = 97) of the total genomes. *bla*_OXA-232_ was most frequently carried by ST231 genomes (61.85%, *n* = 60). Besides ST231, ST2096 (12.37%, *n* = 12) and ST14 (8.24%, *n* = 8) genomes also carried *bla*_OXA-232_ and the remaining 18% of *bla*_OXA-232_ were carried by ST147, ST395, ST23, and ST437 genomes. Four variants of *bla*_NDM_ were detected, namely, *bla*_NDM-1_, *bla*_NDM-4_, *bla*_NDM-5_, and *bla*_NDM-7_. Among these, *bla*_NDM-5_ was observed to be most prevalent (18.18%, *n* = 38), followed by *bla*_NDM-1_ (5.74%, *n* = 12). *bla*_NDM-4_ and *bla*_NDM-7_ were each found only in a single isolate, belonging to ST11 and ST711, respectively. A total of 28.94% (*n* = 11) of *bla*_NDM-5_ was carried by ST147, one of the clinically prevalent STs found in India. Besides this, *bla*_NDM-5_ was also detected in other clinically less prevalent STs: ST395, ST437, ST16, ST14, and ST101. *bla*_NDM-1_ was detected mainly in ST14 and ST11. Single isolates of ST231, ST147, ST273, ST624, and ST2816 also carried *bla*_NDM-1_, whereas ST231 genomes were not found to carry any *bla*_NDM_ variants except for one genome with *bla*_NDM-1_. *bla*_KPC-2_ was also found to be present only in 4 genomes of ST147 and 1 genome of ST101. *bla*_NDM_ and *bla*_KPC_ were not observed to be as prevalent as *bla*_OXA-232_ in India. *bla*_OXA-181_ (11.48%, *n* = 24) was found mainly in ST43, ST147, and ST16. *bla*_OXA-48_ was found in only 2 genomes of ST101. Five different combinations of dual carabapenemase genes were also detected in a few genomes. *bla*_NDM-5_ + *bla*_OXA-232_ was detected in ST147 (*n* = 4), ST437 (*n* = 3), ST2096 (*n* = 2), ST395 (*n* = 2), and ST14 (*n* = 1), *bla_NDM-5_ + bla_OXA-181_* was detected in ST16 (*n* = 5) and ST147 (*n* = 4), *bla*_NDM-5_* + bla*_OXA-48_ was detected in ST101 (*n* = 2), *bla*_NDM-1_ + *bla*_OXA-232_ was detected in ST14 (*n* = 3), and *bla*_NDM-1_ + *bla*_OXA-181_ was seen in ST14 (*n* = 1), ST11 (*n* = 1), and ST42 (*n* = 1) (Figure 1).

Regarding the location of carbapenemase genes, all *bla*_OXA-232_ genes were located on plasmids except 2 genomes (EN5338 and DGL15). Two copies of the *bla*_OXA-232_ gene were detected in *n* = 2 genomes (C18 and SBS3). In *n* = 2 genomes (B35725 and SBS12), *bla*_OXA-181_ was located on the chromosome, while the rest were detected on plasmids. All *bla*_NDM-5_ were detected on plasmids among genomes except 2 genomes (B35725 and SBS12). *Bla*_OXA-48_ and *bla*_KPC_ were present on plasmids in all genomes. All *bla*_NDM-1_ was detected on plasmids, while one genome (JNM8C2) carried 3 copies of *bla*_NDM-1_ on the plasmid.

### 2.3. Distribution of Extended-Spectrum β-Lactamases

Among various ESBLs, *bla*_CTX-M-15_ (80.38%, *n* = 168) was found to be the most abundantly present ESBL. The majority of ST231, ST147, ST2096, ST14, ST16, ST43, ST395, ST11, ST15, ST23, ST45, ST48, ST307, and ST437 genomes carried *bla*_CTX-M-15_. Only the less prevalent STs such as ST515, ST420, and ST101 did not carry *bla*_CTX-M-15_. *bla*_CTX-M-15_ was detected on both plasmids and chromosomes. In ST16 genomes, *bla*_CTX-M-15_ was located on plasmids, whereas in ST231 (*n* = 9), two copies of *bla*_CTX-M-15_ were located on the chromosomes. Further, one genome of ST231 (B28484) had three copies of *bla*_CTX-M-15_ located on the chromosome. A total of *n* = 4 genomes belonged to ST15, ST23, and ST152 carried a double copy of *bla*_CTX-M-15_ gene on their plasmids. Co-existence of *bla*_CTX-M-15_ and *bla*_OXA-232_ was observed in 39.7% (*n* = 83) of genomes. Mainly, genomes of ST231 (77.14%, *n* = 54/70) followed by ST2096 (66.66%, *n* = 10/15), ST14 (50%, *n* = 6/12), ST437 (100%, *n* = 3/3), and ST23 (66.66%, *n* = 2/3) exhibited co-existence of *bla*_CTX-M-15_ and *bla*_OXA-232_. Apart from *bla*_CTX-M-15_, another ESBL, *bla*_OXA-1_ (28.70%, *n* = 60), was also found to be predominant in genomes. *bla*_OXA-1_ was not detected in genomes of the prevalent ST231, and it was detected highest in ST2096 (21.66%, *n* = 13), followed by ST14 (18.33%, *n* = 11), ST147 (10%, *n* = 6), and ST395 (10%, *n* = 6). *Bla*_OXA-1_ was also detected in a few genomes of other prevalent STs, such as ST16, ST11, ST15, ST35, ST48, and ST307. One genome each of ST2096, ST48, and ST11, and three genomes of ST14 had the *bla*_OXA-1_ gene on the chromosome, while the rest was present on plasmids among STs.

### 2.4. Distribution of Broad-Spectrum β-Lactamases

For broad-spectrum *β*-lactamase (BSBL), *bla*_TEM-1_ (74.16%, *n* = 155) was most frequently found in ST231 (42.58%, *n* = 66), followed by ST147 (9.03%, *n* = 14), ST2096 (7.74%, *n* = 12), ST14 (5.80%, *n* = 9), ST16 (4.51%, *n* = 7), ST43 (3.87%, *n* = 6), and ST395 (3.22%, *n* = 5). The co-existence of *bla*_OXA-232_ with *bla*_TEM-1_ was found in 41.14% (*n* = 86) of genomes among different STs, while 65.55% (*n* = 137) of genomes were found to carry both *bla*_CTX-M-15_ and *bla*_TEM-1_. Co-occurrence of these three genes (carbapenemase—*bla*_OXA-232_, ESBLs—*bla*_CTX-M-15_, and BSBLs—*bla*_TEM-1_) was detected in 37.79% (*n* = 79) of the genomes. All *bla*_TEM-1_ genes among genomes were detected on plasmids, except 2 genomes each of ST231 and ST43. Apart from *bla*_TEM-1_, two other BSBLs, *bla*_SHV-1_ (37.79%, *n* = 79) and *bla*_SHV-11_ (25.35%, *n* = 53), were found to be the most circulating across genomes. *bla*_SHV-1_ was majorly carried by genomes of ST231 (72.15%, *n* = 57), followed by ST16 (5.06%, *n* = 4), ST101 (3.79%, *n* = 3), ST515 (3.79%, *n* = 3), and ST48 (2.53%, *n* = 2). Interestingly, we did not find any *bla*_SHV-1_ in the genomes of other prevalent STs, such as ST147, ST2096, ST14, ST43, ST395, ST11, ST15, ST23, ST35, ST307, and ST437. All *bla*_SHV-1_ genes were found to be located on chromosomes, except one genome (BA39950) of ST16 and one genome (BA33875) of ST101 detected with two copies of this gene, where one was on the chromosome while the other was on the plasmid. *bla*_SHV-11_ was detected mainly in ST147 (39.62%, *n* = 21), followed by ST43 (13.20%, *n* = 7) and ST395 (7.54%, *n* = 4). Apart from this, some other prevalent STs, such as ST11, ST23, and ST437, also carried *bla*_SHV-11_ in their genome, and notably the most prevalent ST231, along with ST2096, ST14, and ST16, did not carry the *bla*_SHV-11_ gene, except one genome of ST14 and ST16. *bla*_SHV-11_ was found to be located on chromosomes, except DGL12 (ST16), while MRK9 had two copies of *bla*_SHV-11_, of which one was present on the chromosome while the other was detected on the plasmid.

### 2.5. Other β-Lactamases

Apart from the most frequent carbapenemases, ESBLs, and BSBLs, some other *β*-lactamases such as *bla*_ampH_, which is a penicillin-binding protein that is related to *AmpC,* were found highest (85.16%, *n* = 176) across genomes, and all *bla*_ampH_ genes were located on chromosomes. *bla*_SHV-28_ (13.39%, *n* = 28) was detected in ST2096, ST14, ST15, and ST307. All *bla*_SHV-28_ were present on chromosomes, except one genome (B32205) of ST14. Variants of *bla*_SHV_ (*bla*_SHV-5_, *bla*_SHV-12_, *bla*_SHV-25_, *bla*_SHV-27_, *bla*_SHV-31_, *bla*_SHV-33_, *bla*_SHV-36_, *bla*_SHV-60_, *bla*_SHV-71_, *bla*_SHV-75_, *bla*_SHV-187_), *bla*_CTX-M_ (*bla*_CTX-M-163_, *bla*_CTX-M-238_), *bla*_TEM_ (*bla*_TEM-214_, *bla*_TEM-243_), and *bla*_CMY_ (*bla*_CMY-4_, *bla*_CMY-6_, *bla*_CMY-59_), along with *bla*_OXA-9_, *bla*_DHA-1_, and *bla*_LAP-2_, were also found in a few genomes.

### 2.6. Plasmid Replicons among Dominant STs

A diverse set of plasmid combinations were found to be circulating in Indian genomes and different plasmid combinations were observed to be associated with specific STs (Figure 1 and Figure 5). For instance, most ST231 genomes carried Col440I and ColKP3, while few genomes of ST231 carried ColRNAI. ColRNAI was mainly associated with ST2096 and ST23. ST14 and ST2096 mainly carried ColKP3. Col(BS512) was abundantly present in ST147 genomes, while not frequently carried by ST231, ST14, and ST2096. ColpVC was seen to be associated mainly with ST43 and little with ST147 and ST437. Among prevalent STs, Col440II was carried by ST16, ST23, and ST101. Regarding IncF plasmids, different sets of IncF plasmids were observed to be associated with different STs. IncFIA, IncFIB(pQil), IncFII(K), and IncF(pAMA1167-NDM-5) were abundantly present in ST231 genomes, whereas ST14 harbored only IncFII(K) and IncFIB(K), which were not carried by ST231. A few STs: ST395, ST147, ST43, and ST11, were found to be associated with IncFIB(pQil). IncHI1B(pNDM-MAR) was carried mainly by ST2096, ST14, and ST43. IncFII was carried by the genomes of ST147, ST395, ST14, and ST16. IncFIB(pNDM-Mar) was detected mainly in ST2096 and ST14. IncR was observed to be strongly associated with ST147, while IncFIB(pKPHS1), IncFII(pKPX1), and ColpVC were also detected in genomes of ST147 (Figure 1 and Figure 5).

## 3. Discussion

*β*-lactams are widely prescribed antibiotics for treating *Klebsiella* infections, and carbapenems are one of the last-resort drugs used to treat highly resistant strains. Public health is currently under immediate threat from the advent of Kp that is resistant to carbapenems. To track infections and resistance quickly and affordably, WGS is being employed more and more in research and public health labs. In the present study, 72.24% (*n* = 151) of isolates were detected with at least one carbapenemase gene (either *bla*_OXA_ or *bla*_NDM_), and 19.20% (*n* = 29) of isolates were detected with dual carbapenemase genes in their genome. The two major types of carbapenemase genes, namely *bla*_OXA-48-like_ and *bla*_NDM1/5_, were detected, and these were mainly associated with ST231 and ST147, respectively. To date (9 February 2023), 48 variants of *bla*_OXA-48-like_ and 48 variants of *bla*_NDM_ have been reported (https://www.ncbi.nlm.nih.gov/pathogens/refgene, accessed on 9 February 2023) so far, but fortunately only 4 variants of *bla*_NDM_ and 3 variants of *bla*_OXA-48-like_ were detected in Kp genomes circulating in India.

Recently, Kp genomic surveillance studies from India were reported mostly from South India [15,16]. To the best of our knowledge, in this report, *Klebsiella* genomes from Western India were included for the first time in a surveillance study. Our finding regarding the most common STs, ST231, followed by ST147, corroborates with these studies; however, the third most common ST we found was ST2096, while it was ST14 in the earlier studies. Next, we found *bla*_OXA-232_ as the most prevalent carbapenemase and it was mostly associated with ST231. Similar observations have been reported from South and North India [15,17]. The rapid dissemination of *bla*_OXA-232_ in ST231 can be correlated with the diverse set of mobile genetic elements found neighboring *bla*_OXA-232/181_, which include numerous insertion sequences and transposons of the Tn3 family [18].

*bla*_OXA-232_ was first identified in Kp and *E. coli* isolated from three patients who had been transported from India to France in 2011 [19]. Since then, outbreaks of *bla*_OXA-232_ Kp have been reported worldwide and diverse STs have been identified, including ST14 and ST15 in China [20], ST16 in Thailand [21], ST147 in Germany [22], ST231 and ST2096 in France [23], ST307 and ST101 in The Netherlands [24], and ST437 and ST395 in India. For *bla*_OXA-232_ and *bla*_OXA-181_, both belonging to the *bla*_OXA-48-like_ group, we found 24 genomes that contained *bla*_OXA-181_ and 2 genomes of ST101 with *bla*_OXA-48_. *bla*_OXA-48_-producing Kp from Europe and Africa have been found to belong to ST395 [19]. *bla*_OXA-181_ is currently considered the second most common global *bla*_OXA-48-like_ derivative after *bla*_OXA-48_ [25]. Until 2007, *bla*_OXA-181_ was considered endemic in India as it was reported as the most common *bla*_OXA-48-like_ carbapenemase. However, it is possible that *bla*_OXA-181_ was misreported because of a biased pool of samples from a few centers that had molecular diagnosis facilities. From the current scenario, it is evident that Kp with *bla*_OXA-232_ and *bla*_OXA-181_ are endemic in India with a higher prevalence of *bla*_OXA-232_ (as observed in the present study). *bla*_OXA-48_ is currently the most common *bla*_OXA-48-like_ enzyme globally, followed by *bla*_OXA-181_ [25]. It is interesting to find that though *bla*_OXA-181_ was first reported in India, its prevalence is currently less compared to *bla*_OXA-232_. *bla*_OXA-232_ differs from *bla*_OXA-181_ by a single amino acid substitution, and the genetic environment surrounding the *bla*_OXA-232_ was initially very similar to the environment surrounding *bla*_OXA-181_ [26]. The similarities of the genes, transposons, and plasmids between *bla*_OXA-181_ and *bla*_OXA-232_ suggested a common origin and transposition followed by the subsequent evolution of *bla*_OXA-232_ from *bla*_OXA-181_. However, in the last decade, the genetic environment of *bla*_OXA-232_ (especially from India) has attained vast diversity, as suggested by the MGEs found associated with it [18]. Future studies using long-read sequencing are warranted for a greater number of isolates from India to give a detailed understanding of the exchanges occurring that lead to the successful dissemination of *bla*_OXA-232_ (particularly in ST231) and not *bla*_OXA-181/48_ in India.

Another important observation in the present study was the selected number (*n* = 29) of isolates that co-harbored *bla*_OXA-48-like_ and *bla*_NDM-1/5_. The combination of *bla*_NDM-5_ + *bla*_OXA-232_ was highest (*n* = 11) among 29 dual-producers. Only 2 genomes (B35725 and SBS12) of ST147 with the *bla*_OXA-181_ + *bla*_NDM-5_ combination were found to carry both carbapenemase genes on the chromosome, and the rest of the dual-producers had both genes on plasmids. Genomes of ST16 in this study with *bla*_OXA-181_ were seen to co-exist with *bla*_NDM-5_. Dual carbapenemase producers have also been reported in South Korea [27], Italy [28], Saudi Arabia [29], Iran [30], and Algeria [31]. *bla*_NDM-5_ was mostly carried by ST147 genomes that belonged to bloodstream infections from Tamil Nadu, while other genomes of ST147 were found to carry *bla*_KPC-2_ that belonged to respiratory tract infections from West Bengal. The plasmid replicon IncFIB(K)(pCAV1099-114) was present only in those *bla*_KPC-2_-producing isolates of ST147. A total of 65.78% (*n* = 25/38) of *bla*_NDM-5_ co-existed with *bla*_OXA-48-like_. Patients from South Korea, The United States, and Nepal who had traveled to India or the Indian subcontinent were reported to have *bla*_OXA-48-like_ and *bla*_NDM-5_ [27,32,33]. The presence of carbapenemase duplex (*bla*_OXA-48-like_ and *bla*_NDM-1/5_) among the genomes could lead to pan-carbapenem-resistant isolates, and hence conditions leading to the origination of such duplexes need to be addressed for tackling them. A group evaluated the worldwide spread and genotype distribution of human clinical isolates of *bla*_NDM_-producing Kp and found that *bla*_NDM_ was present in all 5 continents and dispersed among numerous STs [34]. The lack of any dominating lineages suggests that there are not any *bla*_NDM_-positive Kp clones that are obviously high risk. *bla*_NDM_-positive Kp strains are frequently reported to be associated with ST14 [35,36,37]. Another prevalent sequence type in many investigations is ST11 [35,36]. It should be noted that ST11, which is the most common ST of carbapenem-resistant Kp in China, mostly carries *bla*_KPC-2_ rather than *bla*_NDM_ [38]. Even though there is not enough evidence to prove that ST11, ST14, ST15, and ST147 are epidemic clones that are mediating the global spread of *bla*_NDM_, their prevalence across several nations calls for more research.

Although *bla*_NDM_ has been found on bacterial chromosomes [14,39], the majority of carriage occurs on plasmids, which are essential for dissemination. Several different plasmid replicon types have been identified to carry *bla*_NDM_, and the *Enterobacteriaceae* include 20 different replicon types of *bla_NDM_*-carrying plasmids, including the IncC, IncB/O/K/Z, IncFIA, IncFIB, IncFIC, IncFIII, IncHI1, IncHI2, IncHI3, IncN, IncN2, IncL/M, IncP, IncR, IncT, IncX1, IncX3, IncX4, IncY, and ColE10 types [25,37,39]. This indicates that different plasmids have acquired *bla*_NDM_ on several occasions, and it also emphasizes the worrisome fact that many different plasmids are involved in the horizontal transfer of *bla*_NDM_. We also found ambiguity regarding the location of *bla*_CTX-M-15_ among genomes that is still unclear, while other *β*-lactamase genes were dominantly found on either the plasmid or chromosome.

There are a few shortcomings of this study; first, the antibiotic susceptibility data of publicly available genomes were not included for the analysis as the data were not available; second, this is a biased population of isolates that were randomly selected by the respective research/clinical lab, and third, majority of the isolates belonged to South India, especially Tamil Nadu. Hence, there is a need for genome-based surveillance from other parts of the country. Further, the exact correlation as well as copy number of *bla* genes with respective plasmids/chromosomes can only be achieved by long-read sequencing, and this is warranted for future work.

## 4. Materials and Methods

### 4.1. Sample Collection, Identification, and Antibiotic Susceptibility Testing

Clinical samples (*n* = 65) were collected from various pathology labs of Gujarat, India. This included samples from urine, blood, sputum, stool, broncho-alveolar fluid (BAL), wound swab, and endotracheal aspirate (ET). The minimum inhibitory concentration (MIC) of colistin and tigecycline was performed using the broth dilution method. Resistance against other antibiotic classes (penicillins, cephalosporins, carbapenems, monobactams, fluoroquinolones, aminoglycosides, tetracyclines, phenicols, and inhibitors of the folic acid pathway) was detected using the disk diffusion method. Results were interpreted as per Clinical and Laboratory Standards Institute guidelines [40] (data provided in Appendix A). Isolates were then classified as MDR, XDR, and PDR (Table 1) as per the classification criteria reported by Magiorakos et al. [11]. The criteria for defining MDR, XDR, and PDR are as follows: MDR: non-susceptible to >1 agent in >3 antimicrobial categories, XDR: non-susceptible to >1 agent in all but <2 categories, i.e., isolates remain susceptible to only one or two categories, and PDR: non-susceptible to all antimicrobial agents [11]. To explore the resistome and plasmidome of these resistant isolates from all 3 resistant categories, we performed whole genome sequencing (WGS).

### 4.2. Genomic DNA Extraction, NGS Library Preparation, and Whole Genome Sequencing

Genomic DNA for 22 isolates of Kp was extracted using the XpressDNA Bacterial kit (MagGenome, Chennai, India). Whole genome sequencing of 6 isolates (*SBS1, SBS3, SBS4, SBS5, SBS11,* and *SBS12*) was performed using the Ion Torrent (S5-0083-GGI) NGS platform (ThermoFisher Scientific, Waltham, MA, USA), and another 16 isolates (*DGL1, DGL2, DGL5, DGL6, DGL7, DGL8, DGL9, DGL10, DGL11, DGL12, DGL13, DGL14, DGL15, DGL16, DGL17,* and *DGL18*) were sequenced using the Illumina MiSeq platform (Illumina, CA, USA). The Ion Xpress™ Plus gDNA fragment library preparation kit (ThermoFisher Scientific, MA, USA) and the Nextera XT DNA Library Prep Kit (Illumina, San Diego, CA, USA) were used to prepare NGS libraries, respectively, as per the instructions in the manuals. The sequences were submitted to NCBI, and the accession number for each sequence is included in Table 1.

### 4.3. Analysis of Whole Genome Sequencing Data

Raw data were analyzed using FastQC (https://www.bioinformatics.babraham.ac.uk/projects/fastqc/, accessed on 2 December 2020) and filtered using the FastX toolkit (http://hannonlab.cshl.edu/fastx_toolkit/, accessed on 2 December 2020). Filtered reads were assembled de novo using SPAdes v3.14.1 (http://cab.spbu.ru/software/spades/, accessed on 4 December 2020) with read error correction and assembling mode and a threads value of 32. Quality assessment of assembled sequences was performed using QUAST v5.0.2 (http://quast.sourceforge.net/quast, accessed on 5 December 2020). Assembled genomes were annotated using Prokka v1.13.3 (https://github.com/tseemann/prokka, accessed on 7 December 2020). The assembly data were deposited on NCBI (Bioproject No: PRJNA694019)

### 4.4. Retrieval and Analysis of Publicly Available Genomes of Klebsiella Pneumoniae

Publicly available genome sequences of MDR Kp (*n* = 187) deposited from India were downloaded from the PATRIC database (https://www.patricbrc.org/, accessed on 25 September 2021). To access genomes deposited from India with antimicrobial resistance properties, filters such as ‘India’, ‘Antimicrobial resistance’, and ‘Genome quality: good’ were used. Recently, in December 2022, this database was merged with the Bacterial and Viral Bioinformatics Resource Center (https://www.bv-brc.org/, accessed on 8 February 2023). The sequences in FASTA format, which were previously assembled and had good quality, were retrieved and used for the combination analysis with our samples. Strains’ information such as collection year, geographic location, sample origin, and host health for each genome was also collected from PATRIC.

MLST profiles for all isolates in the PATRIC database were mentioned in the metadata, except for a few isolates, and sequence types for our (*n* = 22) isolates and those remaining from the PATRIC database were determined using MLST v2.0.4 of the Centre for Genomic Epidemiology (CGE) toolbox (https://cge.cbs.dtu.dk/services/MLST/, accessed on 17 November 2021). For detection of STs, 7 housekeeping genes (*gapA*, *infB*, *mdh*, *pgi*, *phoE*, *rpoB*, *tonB*) and their allelic combinations were used to generate STs. All isolates (*n* = 209) were analyzed to detect the *β*-lactamase genes as well as the copy numbers (analyzing ORF_ID, Contig, and Start and Stop position in genomes) using the Resistance Gene Identifier (RGI) from The Comprehensive Antibiotic Resistance Database (CARD) (https://card.mcmaster.ca/analyze/rgi, accessed on 17 November 2021). The location of *β*-lactamase genes in the genome was detected using mlplasmids v2.1.0 (https://sarredondo.shinyapps.io/mlplasmids/, accessed on 5 December 2022) (Arredondo-Alonso et al., 2018). Plasmids were detected using PlasmidFinder v2.0.1 of the CGE toolbox (https://cge.cbs.dtu.dk/services/PlasmidFinder/, accessed on 15 December 2021). The single-nucleotide polymorphism (SNP)-based phylogenetic tree was generated using CSI Phylogeny 1.4, and MGH78578 was used as a reference strain. iTOL v6 (https://itol.embl.de/, accessed on 10 February 2023) was used to visualize the phylogeny and genomic profiles of isolates.

## 5. Conclusions

Collectively, this is the first surveillance analysis of carbapenem-resistant Kp Pan-India genomes, which includes WGS of 194 clinical, 14 environmental, and 1 unknown strain that were collected from various geographic locations of India between 2012 and early 2021. In this surveillance analysis of MDR Kp circulating in India, ST231 was found to be the most predominant ST. We also identified one novel ST5438 in our isolates. In regards to carbapenemase genes, *bla*_OXA-232_ was the most circulating, followed by *bla*_NDM-5_, while *bla*_CTX-M-15_ was highest among ESBLs followed by *bla*_SHV-1_, and these genes can be targeted for diagnostic purposes. IncFII(K) was the most frequent plasmid replicon belonging to the Inc type, while ColKP3 was detected as the second most prevalent among Col-type plasmids. ST147 is already known as a high-risk clone globally, while ST16 and ST14 from this study, which are mostly dual-producers of carbapenem and ESBL genes, could be emerging high-risk clones in India. Our study suggests an unmet need of future large-scale, multi-regional genomic surveillance of multidrug-resistant Kp isolates with collaboration across different states of India, especially from Northern and Western India.

## Figures and Tables

**Figure 1 antibiotics-12-00449-f001:**
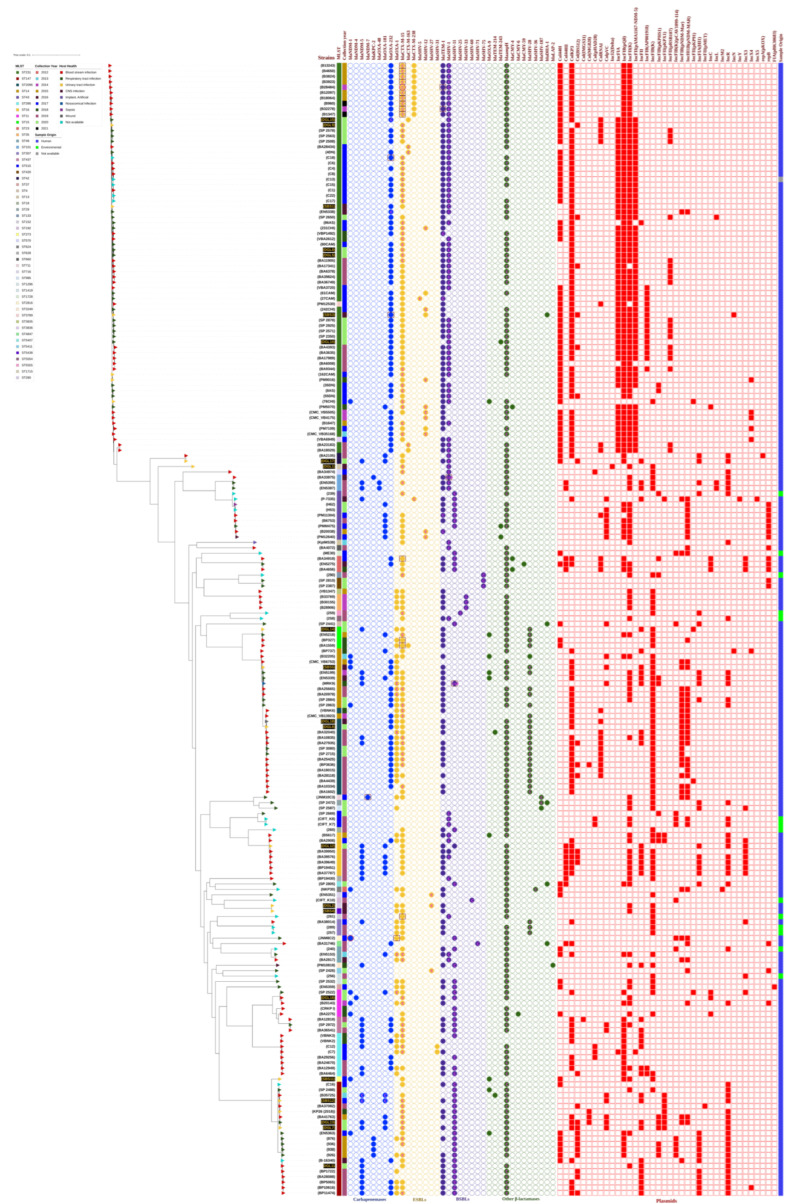
Genomic analysis of *β*-lactamase genes and plasmid replicons circulating in Kp genomes from India. Strains in yellow denote the lab isolates, black denotes retrieved genomes. Circles with filled color (blue—carbapenemase, yellow—ESBLs, purple—BSBLs, and green—other *β*-lactamases) for beta-lactamase genes denote the presence of genes, unfilled denotes the absence of genes. C inside a circle denotes the location of genes on the chromosome, without C denotes genes on plasmids. Red ring outside the circle denotes 3 copies of the gene, dark red square outside the circle denotes 2 copies of the gene. C and P denote that one is on chromosome and another is on the plasmid. For plasmids, red color-filled square denotes the presence of a plasmid, unfilled denotes the absence. iTOL was used to create this image.

**Figure 2 antibiotics-12-00449-f002:**
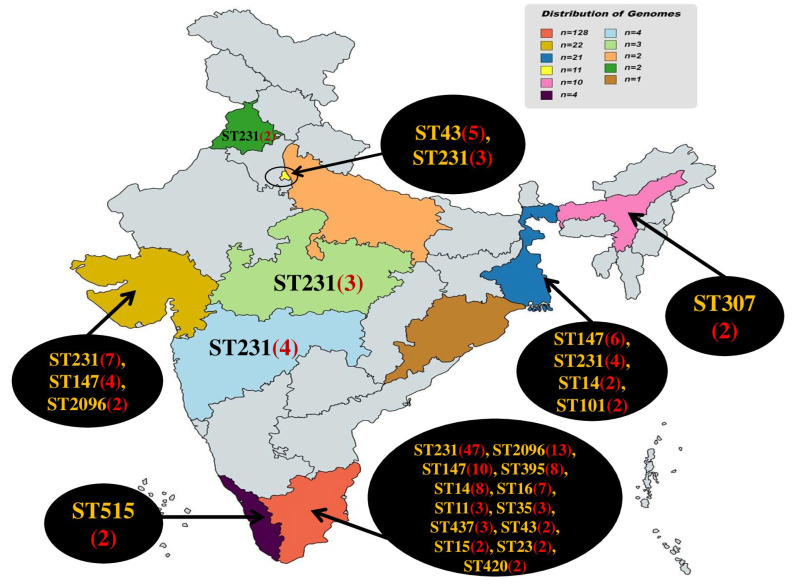
Geographic location of genome collection and state-wise ST distribution. ST found in at least 2 genomes are included in this figure. (*n*) Represents the number of genomes, and colors represent the states: carrot red (*n* = 128)—Tamil Nadu; mustard yellow (*n* = 22)—Gujarat; cyan (*n* = 21)—West Bengal; yellow (*n* = 11)—New Delhi; pink (*n* = 10)—Assam; purple (*n* = 4)—Kerala; sky blue (*n* = 4)—Maharashtra; pastel green (*n* = 3)—Madhya Pradesh; orange (*n* = 2)—Uttar Pradesh; green (*n* = 2)—Punjab; brown (*n* = 1)—Orrisa. Digits in red fonts in brackets denote the number of genomes having respective STs. MapChart online tool was used to generate this figure.

**Figure 3 antibiotics-12-00449-f003:**
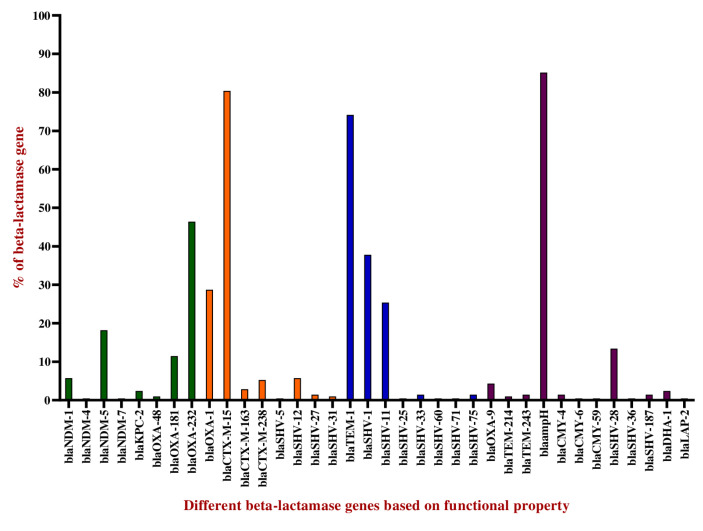
Percentage distribution of *β*-lactamase genes among strains. Bars in the graph indicate individual *β*-lactamase genes, in which bars in green indicate the carbapenemase gene, orange indicates ESBLs, blue indicates BSBLs, and purple indicates other *β*-lactamase genes. The GraphPad Prism 8.4.2 tool was used to create this graph.

**Figure 4 antibiotics-12-00449-f004:**
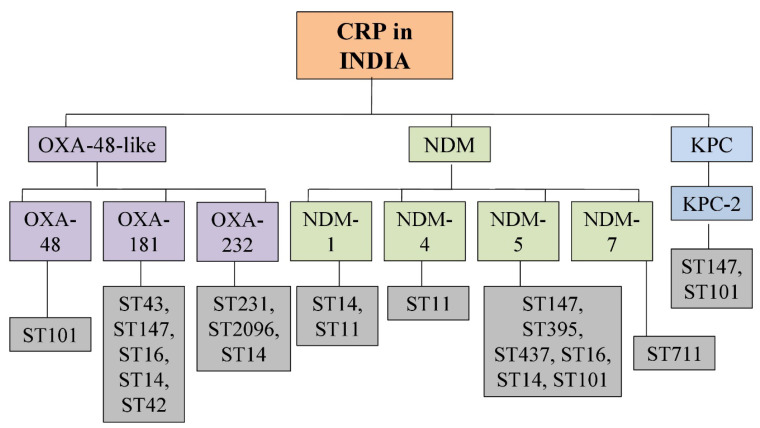
Distribution of carbapenemase genes circulating in India. CRP—carbapenem-resistant pattern.

**Figure 5 antibiotics-12-00449-f005:**
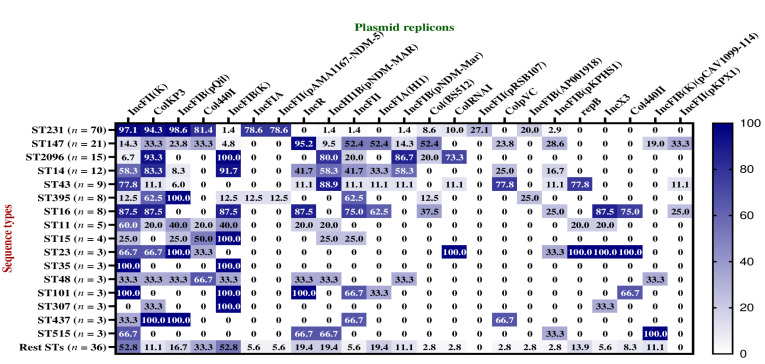
Heatmap of percentage distribution of plasmid replicons among STs. (*n*) Indicates the total number of particular STs. On the scale, dark blue indicates 100%, white indicates 0%. The GraphPad Prism 8.4.2 tool was used to generate this heatmap.

**Table 1 antibiotics-12-00449-t001:** List of isolates (strain) with resistance category and genome accession number submitted in NCBI (Bioproject no: PRJNA694019).

Strain	Resistance Category	Genome Accession No.	Strain	Resistance Category	Genome Accession No.
*SBS1*	XDR	JAFFRJ000000000	*DGL8*	PDR	JAJBAG000000000
*SBS3*	XDR	JAFFRI000000000	*DGL9*	XDR	JAJBAF000000000
*SBS4*	MDR	JAFFRH000000000	*DGL10*	XDR	JAJBAE000000000
*SBS5*	XDR	JAFFRG000000000	*DGL11*	XDR	JAJBAD000000000
*SBS11*	MDR	JAFFRA000000000	*DGL12*	XDR	JAJBAC000000000
*SBS12*	PDR	JAFFQZ000000000	*DGL13*	XDR	JAJBAB000000000
*DGL1*	MDR	JAJBAN000000000	*DGL14*	XDR	JAJBAA000000000
*DGL2*	MDR	JAJBAM000000000	*DGL15*	XDR	JAJAZZ000000000
*DGL5*	PDR	JAJBAJ000000000	*DGL16*	XDR	JAJAZY000000000
*DGL6*	XDR	JAJBAI000000000	*DGL17*	XDR	JAJAZX000000000
*DGL7*	XDR	JAJBAH000000000	*DGL18*	XDR	JAJAZW000000000

## Data Availability

Data will be available upon request.

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
