# Peer review of "Diversity and Distribution of β-Lactamase Genes Circulating in Indian Isolates of Multidrug-Resistant Klebsiella pneumoniae"

_antibiotics, 2023, doi:10.3390/antibiotics12030449_

Round 1
Reviewer 1 Report
The manuscript by Shukla et al. describes the molecular, phenotypic and genomic analysis of MDR Klebsiella pneumoniae isolates circulating in India with a focus on beta-lactamase genes. In general, the data presented is interesting and the analysis is thorough. However, I found the manuscript structure messy and hart to follow. I suggest making the revision before the paper can be accepted for publication. I have several major and minor comments for the authors.
Major comments:
1. Results section is very hard to follow. I suggest adding separate sections for STs, ESBL types, geographic distribution and plasmids and distribute the figures between the appropriate sections. Currently, all figures are place at the end, which makes the reading very uncomfortable.
2. Please specify how did you investigate the resistance to colistin – did you perform serial broth dilutions as required? Taking just one MIC threshold is not considered sufficient
3. Why did you use a more than one-year old data available rom PATRIC? Currently there are almost 500 genomes from India available in this database – in the field of WGS the timely updates of data are crucial since the amount of available genomes grows exponentially
Minor comments:
Title (and throughout the text) – more commonly used term is “multidrug-resistant”, not “multi-drug resistant”, and this form is also used in some parts of the text – please use this term consistently
Line 24 – “from various pathology lab” – probably, should be “labs”
Line 25 – “for (n=22) resistant isolates” – please specify here whether these isolates were MDR, XDR or simply resistant to several classes of antimicrobials
Line 28 – “location” – did you mean resistance gene location, e.g., chromosomal or plasmid? It is not clear
Line 31 – correct gene name should be “ampH”, not “AmpH”
Lines 32,33 – “ was observed to prevalent” – should be “was observed to be prevalent” “was observed to prevail”
Line 36 – “ ST16 &ST14 from this study, which is…producer” – should probably be “are… producers”
Line 44 – please define GI here
Line 49 – should be “IncX3”
Lines 52-54 – references to websites should be moved to the reference list and linked appropriately; the dates of last access to these pages should be specified (see journal instructions for authors)
Line 63 – “For e.g. Greece” - for is redundant. Should be “E.g.,” or “For example”
Line 63-65 – Firstly, you state that carbapenem resistance has “little geographic variation” (actually, this is not the case – please provide references) and then you state that there are differences in bla genes frequency between Europe and India – please explain and provide support for your data (reference to papers, databases etc)
Line 77 - “they also reported that tigecycline and colistin are” – should be “they also reported that tigecycline and colistin were” (sequence of tenses)
Results – I suggest specifying not sample identifiers (SAMN), but genome or assembly accessions (GCA or the like) since sample could contain multiple genomes and genome versions could be updated later with new data
Line 104 – “and nidentified as ovel” – please fix typos
Please place Figure 1 closer to its first mention in text. Please increase the resolution – gene names and sample names are hardly distinguishable
Line 249 – “denote number of STs” – not STs, but the number of isolates or genomes having this ST – please fix
Figure 3 – please make the gene names oriented vertically – it is very hard to distinguish them
Table 1 – I suggest using genome accessions, not biosamples for the reasons given above
Line 273 – “that is resistant to carbapenem” – should probably be “to carbapenems”
Line 281 – please specify the date of access to web resource (the number of variants could change over time)
Line 371 – please specify the date of your last access to web resource. Currently, this links gives “page not found” error
Line 393 – please specify spades version and parameter used because this could affect the assemblies
Line 424 – “carabapenem” – please fix typo
All references in reference list have duplicated numbers.
Reviewer 2 Report
Major
· Your results contain more details about gene presence that could be found in tables, or figures, some parts are missing such as results of antimicrobial susceptibility and MIC. Also, you have to compare phenotypic and genotypic resistance because your title is focusing on this issue.
· It's better to study the capsule-type genes, sub-lineages, and identification of hypervirulent and classical strains
· BlaTEM-1 and blaSHV-1 are not ESBLs why did you include them? Also, you mentioned "Genes of BSBLs, blaTEM-1" this is not correct please revise these results in the whole manuscript
Minor
· Keep italics only for the isolates names, not for the abbreviation such as Kp
· Lines 52-54 instead of these long links you can add a citation
· Line 104 spelling mistake " nidentified "
· How did you confirm the presence of multiple copies of the genes?
· The isolates' names are not clear in figure one
· How did you specify the MDR, XDR, etc.. phenomenon because your phenotypic data is missing
·
Round 2
Reviewer 1 Report
My comments have been addressed. I have no further concerns regarding the manuscript.
Reviewer 2 Report
· Line 103 " n=3 XDR isolates (SBS1, SBS3, SBS5, DGL18)" you said n=3 and you included four !
· Line 104, In supp. files SBS1 is sensitive also to Amoxicillin-clavulanic acid in addition to colistin and tigecycline, please check the rest of your results
· Remove the underlines
